# Genes for highly abundant proteins in *Escherichia coli* avoid 5' codons that promote ribosomal initiation

**Loveday E. Lewin**[☯], **Kate G. Daniels**[☯], **Laurence D. Hurst**[ID]*

The Milner Centre for Evolution, Department of Life Sciences, University of Bath, Bath, United Kingdom

☯ These authors contributed equally to this work.
* l.d.hurst@bath.ac.uk

**Data Availability Statement:** The scripts and data necessary to generate all results and figures, including those in Supplementary Information, have been deposited in 10.5281/zenodo.8349580.

## Abstract

In many species highly expressed genes (HEGs) over-employ the synonymous codons that match the more abundant iso-acceptor tRNAs. Bacterial transgene codon randomization experiments report, however, that enrichment with such "translationally optimal" codons has little to no effect on the resultant protein level. By contrast, consistent with the view that ribosomal initiation is rate limiting, synonymous codon usage following the 5' ATG greatly influences protein levels, at least in part by modifying RNA stability. For the design of bacterial transgenes, for simple codon based *in silico* inference of protein levels and for understanding selection on synonymous mutations, it would be valuable to computationally determine initiation optimality (IO) scores for codons for any given species. One attractive approach is to characterize the 5' codon enrichment of HEGs compared with the most lowly expressed genes, just as translational optimality scores of codons have been similarly defined employing the full gene body. Here we determine the viability of this approach employing a unique opportunity: for *Escherichia coli* there is both the most extensive protein abundance data for native genes and a unique large-scale transgene codon randomization experiment enabling objective definition of the 5' codons that cause, rather than just correlate with, high protein abundance (that we equate with initiation optimality, broadly defined). Surprisingly, the 5' ends of native genes that specify highly abundant proteins avoid such initiation optimal codons. We find that this is probably owing to conflicting selection pressures particular to native HEGs, including selection favouring low initiation rates, this potentially enabling high efficiency of ribosomal usage and low noise. While the classical HEG enrichment approach does not work, rendering simple prediction of native protein abundance from 5' codon content futile, we report evidence that initiation optimality scores derived from the transgene experiment may hold relevance for *in silico* transgene design for a broad spectrum of bacteria.

## Author summary

Transgene experiments in *Escherichia coli* report that codon usage in the first few amino acids after the initiating ATG has a profound influence on the resulting protein level by

**Funding:** The author(s) received no specific funding for this work.

**Competing interests:** The authors have declared that no competing interests exist.

promoting ribosomal initiation, rather than enabling speedy elongation. For the design of bacterial transgenes and for simple codon-based *in silico* inference of protein levels, it would be valuable to computationally determine initiation optimality scores for codons for any given species. An attractive approach is to characterize the 5' codon enrichment of highly expressed genes compared with the most lowly expressed genes, just as translational optimality scores of codons (affecting elongation) have been similarly defined employing the full gene body. Using unique resources provided for *E. coli* we show that, unexpectedly, this doesn't work: the 5' ends of highly expressed genes are enriched, compared to lowly expressed ones, in codons that objectively are associated with low initiation rates. This likely reflects conflicting selection pressures in highly expressed genes which can favour low initiation to promote low noise or high efficiency. While simple prediction of native protein abundance from 5' codon content is then somewhat futile, the objective initiation optimality scores may hold relevance for *in silico* transgene design for a broad spectrum of bacteria.

## Introduction

As they do not obviously affect the resultant protein, synonymous mutations were originally assumed to be selectively neutral [1]. As it subsequently became clear that synonymous mutations can be under selection [reviewed in 2,3–5], analysis of the causes of their fitness effects provided a means to determine why the simple null model is wrong, leading to a fuller understanding of proteogenesis [5–7]. This understanding in turn has application to codon-based proxy measures of gene expression level [8], to the design transgenes for biotechnological and medical applications [5,9–15] and for diagnostics [16–21].

Historically, the literature on selection on synonymous sites, and transgene design [9–12,22,23], has been dominated by the concept of translational selection [24–28]. This envisages that selection on codon usage [29] is mediated by effects of tRNA availability [26–28] on translational speed [30] (but see [31]) or accuracy [32,33]. There is then the concept of the "translationally optimal" codon [25], the one within any synonymous codon block matching the most abundant iso-acceptor tRNA [25,34]. This definition in turn informs some metrics of translational optimality [35]. However, given difficulties knowing concentrations of charged iso-acceptor tRNAs, operationally the degree of translational optimality of a codon in any given species is most commonly defined by its degree of enrichment in highly expressed genes (HEGs) compared with lowly expressed genes (LEGs), this providing the basis for the commonly employed codon adaptation index (CAI)[25]. As required by this metric, codons enriched in HEGs tend to specify the more abundant tRNAs [25,34,36]. Notice that codon optimality can be both considered a discontinuous concept whereby for each amino acid there is a unique optimal codon all others being non-optimal, or a continuous variable, each codon being granted a score dependent on its degree of enrichment compared to synonyms in HEGs.

With the continuous measure, the CAI score for a gene as a whole is in turn utilized as a proxy measure of gene expression level [8] and it is commonplace when designing genes for transgenesis to over-employ the high CAI codons [9–12,22,23], under the assumption that this will boost protein levels. A closely related approach employs codons in a manner proportional to their host gene usage [13]. While within the biotechnology literature the assumption that translationally optimal codons *cause* high expression has been "*de rigeur*"[5], whether enriching for high CAI codons promotes protein level expression of a transgene is not as robustly demonstrated as might be expected given the commonality of the practice [9,31,37,38]. With

advances in gene manipulation technology, the causality question has been addressed by large scale synthetic gene (alias transgene) codon randomization experiments in *Escherichia coli* [38–43]. In these, many versions of a gene (or genes) differing at synonymous sites are constructed and the protein level associated with each assayed [38–43]. Depending on the experimental design, all other parameters are controlled either experimentally or statistically allowing the role of codon translational optimization (or other prospective determinants) to be evaluated [38–43].

These experiments report that usage of translationally non-optimal codons (low gene-level CAI) causes decreased cell fitness [38,39, 43], possibly owing to ribosomes being held on unwanted transcripts [38,43]. Importantly, however, they also find that translationally optimal codon usage (high gene-level CAI) does not increase protein level [38,39,41]. Kudla et al. [38], for example, report that the CAI score of otherwise identical GFP transgenes explains only 2% of the between-transgene variation in protein level [38]. While there are concerns that this data set may be biased [43], Nieuwkoop et al [41] similarly find only a very minor role for codon adaptation in determining protein level. Cambray et al. [43] recently clarified further, finding that codon translational optimization can have an effect, but only in rare elongation-limited transcripts. These results accord with the finding that one amino acid one (translationally optimal) codon strategy for transgene design is ineffective [15].

Such rare or weak effects on protein levels suggest that selection on codon translational optimality is not selection to make more of a given protein, but rather to enable more efficient manufacture of all proteins by rapidly freeing initiated ribosomes [5]. This is reasonable as faster processing of a given transcript need not ensure more of that protein, the rapidly released ribosome being free to process other transcripts not the focal one [38,43]. If so, the correlation between CAI and expression level across native genes [29] reflects cellular costs associated with non-optimal usage rather than direct consequences for protein level [5]. This comes with the caveat that there are systems to terminate slowly translated transcripts [44–46] and optimal codons can boost transcription [47–49].

Assuming that CAI does not well predict transgene protein levels, what might then explain the universally reported [38–43] large variation in protein levels when synonymous sites are randomised? All transgene randomization experiments in *Escherichia coli* find that the codon usage of the 5' ends of the coding sequence (CDS) (meaning approximately the first 10 codons) have a profound effect on protein level [38–42]. Nieuwkoop et al [41], for example, employed a Random Forest model to dissect the causes of variation in levels of red fluorescent protein (RFP) that differ at synonymous sites, reporting that the codon composition at the 5' end of the CDS was highly influential, with codon translational adaptation of these not being a predictor. Similarly, Allert et al [42] and Kudla et al [38] both report that high AU content at CDS 5' ends is predictive of protein levels of codon randomised transgenes. These results accord with early comparative genomic evidence that in *E. coli* and *Salmonella typhimurium* the 5' end of CDS tends to be especially slow evolving and have unusual non-synonymous and synonymous site content, often being A rich and G poor [50] (later more broadly confirmed [51]) (see also [52,53]).

The most parsimonious model for this 5' CDS end effect [38–41] supposes that it is mediated in large part by selection for low 5' RNA stability [38,39,42,43,54–57], as hypothesised [50], and not by tRNA pool effects. Indeed, both Goodman et al [39] and Nieuwkoop et al [41] agree that in this 5' domain low RNA stability predicts higher protein levels and that the CAI value of the codons in the 5' end is irrelevant, at least when controlling for stability effects. A boosting effect of low RNA stability can explain the preference for thermodynamically less stable high AU content [42], especially high A content as U enables non-canonical U:G pairing [39]. Note that the term RNA stability in this context, refers exclusively to energetic stability,

commonly measured as Gibbs free energy, of predicted RNA secondary structures and not to RNA degradation or measures of half-life.

There are multiple, not mutually exclusive, mechanisms by which 5' mRNA stability might matter, all of which suggest that low stability of the 5' end affects translational initiation broadly defined [38,39,50], i.e. meaning the chance that the RNA is bound by the ribosome, the ribosome finds the start codon or the translation is effectively progressed without early RNA degradation. As the ribosome has helicase activity [58], it is likely that strong 5' RNA folding is a barrier to such initiation. Strong RNA folding could also force inaccessibility of the upstream Shine-Delgarno (SD) sequence [59] that in turn affects mRNA and protein levels, this explaining why 3' mutations disrupt translation if interacting with the SD sequence and why RNA structures tend to avoid SD interactions [59]. RNA stability may additionally mediate susceptibility of the mRNA to degradation, determined by the extent to which the early mRNA is uncovered by ribosomes [60]. The ramp hypothesis proposes that any RNA stability effects have their benefit through modulating ribosome velocity and efficient queuing [61]. The related suggestion that codon composition also modulates protein level by slowing ribosomes by matching rare tRNAs [61–63] is more controversial [55]. Indeed, when synonymous sites of translationally optimal codons are AT rich, usage of the AT rich synonymous sites still increase protein levels [55]. If generally true, 5' enrichment of non-optimal codons [61] is then a necessary consequence of selection for AT richness *per se*, not for codon non-translational optimality as such.

Just as we can consider codons that match the tRNA pool as translationally optimal (TO) codons, assayed by CAI, so too we can consider those codons that promote this initiation as "Initiation Optimal" (IO) codons. We here are not so much concerned with the mechanism by which the 5' codon effects manifest alterations in protein levels, nor are we concerned with determining which parameters best predict protein expression levels, this being well resolved [43]. Rather we seek to ask whether it is possible to determine from a purely *in silico* analysis initiation optimality scores for every codon for any given bacterial species, just as we routinely attempt to infer translational codon adaptation indexes [64].

As noted above, for translationally optimal codons the common approach is to determine enrichment of codons, compared with synonyms, seen in HEGs compared with LEGs [25,64]. We seek to determine whether the same methodology works when applied to define initiation optimal codons: is the enrichment of codons at the 5' ends of native genes (as opposed to transgenes) specifying the most abundant proteins a good guide to the initiation optimality scores for any given species? If so, then one can simply adapt methodologies for determining CAI for a gene body to a methodology to determine IO scores restricting analysis to the 5' end in native genes.

This enterprise is important in three contexts. First, determining the IO metric for any species can inform transgene design. That TO codons do not majorly affect protein level while IO ones do (see above), provides an immediate rationale for finding a simple method to determine IO codons of any given species. This in part motivates our focus on codon composition (as opposed to, for example general 5' stability) as transgene design algorithms commonly employ codon-level metrics to determine which synonymous site to employ within the transgene, sometimes in a gene position specific manner (e.g. [65–67]). Second, while CAI provides a useful surrogate for expression level of a gene in some species [8,36], understanding IO scores could augment such a metric providing a better proxy of protein abundance, potentially applicable also in species in which there is no evidence for translational selection [36] and no (hard to obtain) protein level measures. A simple metric of mean 5' IO score may then be useful. This again informs our choice to define codon specific IO measures. Third, in a broader context, such an analysis permits us to understand the general utility of employing highly

expressed genes as the end point of a monotonic continuum in determining the direction of selection. This method is employed not just to derive the translationally optimal codons and species-specific CAI [25]. It is also routinely employed to determine whether features of gene expression and gene product diversity [68] might be selectively optimal (see e.g. analyses of alternative translation initiation sites [69], RNA modifications [70], circular transcripts [71], RNA editing [72], alternative polyadenylation sites [73,74] and stop codon read-through [75,76]). In all cases, we assume that the trends seen as we move from low expression to high expression tell us the direction of selective optimality. Some evidence suggests that the methodology has a strong basis. There is, for example, correspondence between tRNA availability and codon usage bias seen in highly expressed genes in many species [25,34,36], and highly expressed genes over-employ the stop codon experimentally determined to be least subject to read-through (i.e. TAA)[76–78]. Asking whether native HEGs are enriched for objectively defined IO codons at 5' ends adds to this literature.

In this context, data for *Escherichia coli* presents a unique opportunity as there is both a large-scale transgene experiment designed to determine the effects of codon modification at the CDS 5' end [39] and, from PaxDB [79], the most complete proteomic coverage of any species. This transgene experimental set [39] is not only designed specifically to address the problem of which 5' codons promote protein production, it is the only data set that permits definition of an initiation optimality score for all synonymous codons (see Methods). As translational optimality does not predict protein levels, but 5' codon usage does, we presume that this experiment is measuring initiation optimality very broadly defined (NB it controls for transcript abundance). Indeed, we consider the transgene experimental data as a providing as near a Gold-standard measure of initiation optimality (broadly defined) for all codons as is currently available (see Discussion for caveats). We then, in turn, ask whether trends of codon enrichment in the 5' ends of native genes specifying high abundance proteins (compared to low abundance proteins) are the same as the codon enrichment trends associated with high transgene protein abundance (controlling for transcript level). This we do by asking whether log odds ratio scores for enrichment at 5' ends in native HEGs is positively correlated with the log odds ratio scores for 5' enrichment in the more abundant proteins in the experimental transgene data.

Most surprisingly, given the history of using enrichment in native HEGs to define codon translational optimality, we find that the 5' codons employed by native HEGs are skewed towards those that have low IO scores (derived from transgenes) that we presume reduce initiation. This strongly suggests that, in contrast to CAI, we cannot–and must not–derive initiation optimality scores from 5' codon usage enrichment in native HEGs. We also cannot infer native protein levels from 5' codon usage using the experimental data as a guide. The result also tempts a series of further questions. First, we ask why native genes of highly abundant proteins use 5' codons that are associated with (presumed) low initiation rates in the experimental data. Second, we ask whether there might be an alternative defensible *in silico* approach to defining initiation optimality for any given bacterial species.

## Results

### A/T ending codons promote high protein levels in *E. coli*

We start by considering the log odds ratio of each codon being observed, compared to the synonyms, in the mRNA 5' domain of the 25% most highly abundant proteins compared with the 25% lowest in the transgene experiment [39] (after normalization–see Methods). Note that this method excludes any effects of skewed amino acid or codon usage in the constructs and controls for transcript level differences. This confirms prior results [39] and indicates that *A*

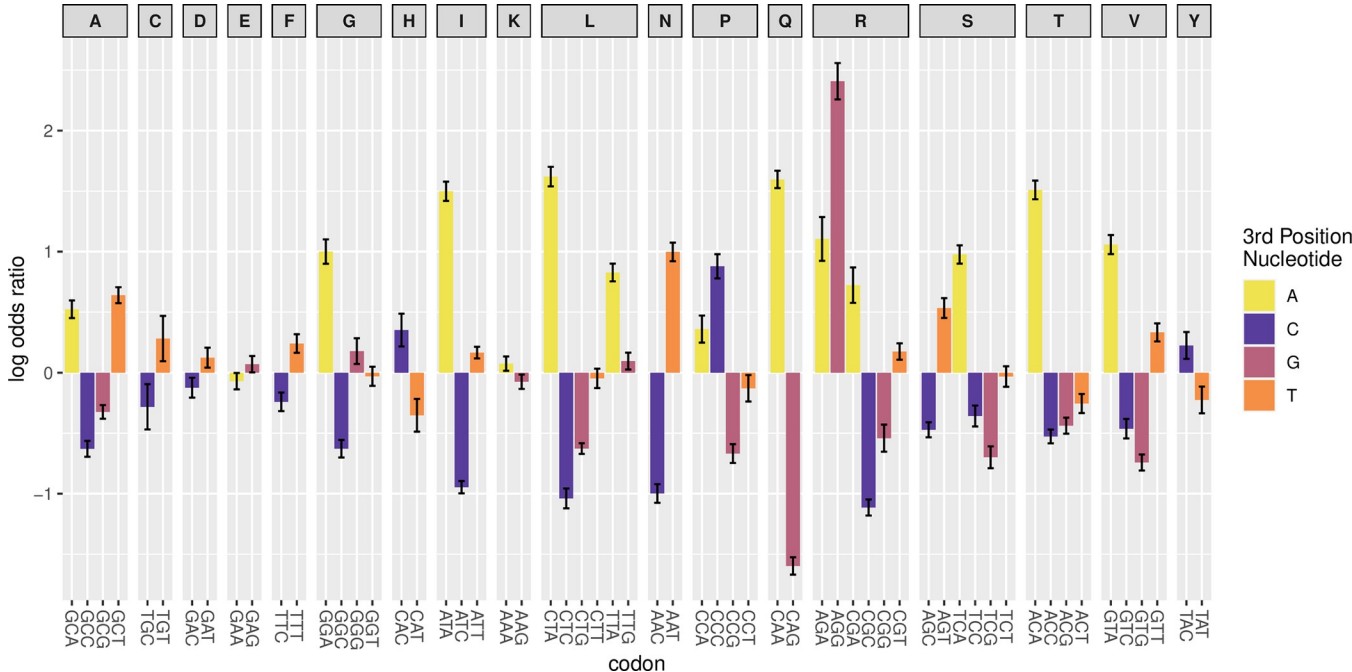

**Fig 1. Log odds ratio for codons associated with high expression when at 5' ends in experimental data.** Error bars are standard errors (see Methods).

ending codons are commonly associated with high expression and, if an *A* ending codon isn't possible, then a *T* ending one is most associated with high protein levels (per transcript) instead (Fig 1 and S1 Table: notice in Fig 1 *A* is coded yellow, *T* orange). Of 30 A or T ending codons, 22 have positive log odds ratios (binomial test, P = 0.016). Conversely, of 29 G or C ending codons, all but 7 have negative log odds ratios (binomial test, P = 0.024).

There are, however, exceptions to the rule that the A/T ending codon is the most associated with high protein level. Of the six arginine codons, AGG has the highest log odds ratio and is, indeed the codon with the highest log odds ratio. For proline, histidine and tyrosine, the *C* ending codon is associated with a greater than zero log odds ratios (CCC, CAC, TAC respectively). The 59 element vector of log odds ratios (S1 Table) we term $V_{edIO}$ (edIO = **e**xperimentally **d**emonstrated **I**nitiation **O**ptimality) and regard as the current Gold standard measure of initiation optimality for *E. coli*.

## In *E. coli* the 5' of genes of the most abundant proteins avoid codons associated with efficient ribosomal initiation

We now ask whether 5' ends of *E. coli*'s native genes associated with the highest protein levels are associated with over-employment of the Gold Standard set of initiation optimal codons. To consider this we derive the log odds ratios of codon enrichment at the 5' end comparing the top 25% by native protein abundance and the bottom 25%. This 59 element vector we term $V_{5-prot}$ (S1 Table). Surprisingly, we find that this vector is anti-correlated with that derived from experimental data (*r* = -0.44, P = 0.0005; Fig 2A), demonstrating that the native genes for the most highly expressed proteins under-employ what we presume to be initiation optimal codons at their 5' ends. The same trend is seen when comparing between synonymous codons within a synonymous codon block, the codon that more enables higher expression in a transgene is the one avoided in the 5' end of the genes for the most abundant proteins (*r* = -0.3, P = 0.005, Fig 2B). These results are robust to control for outliers (for Fig 2A: Spearman rank

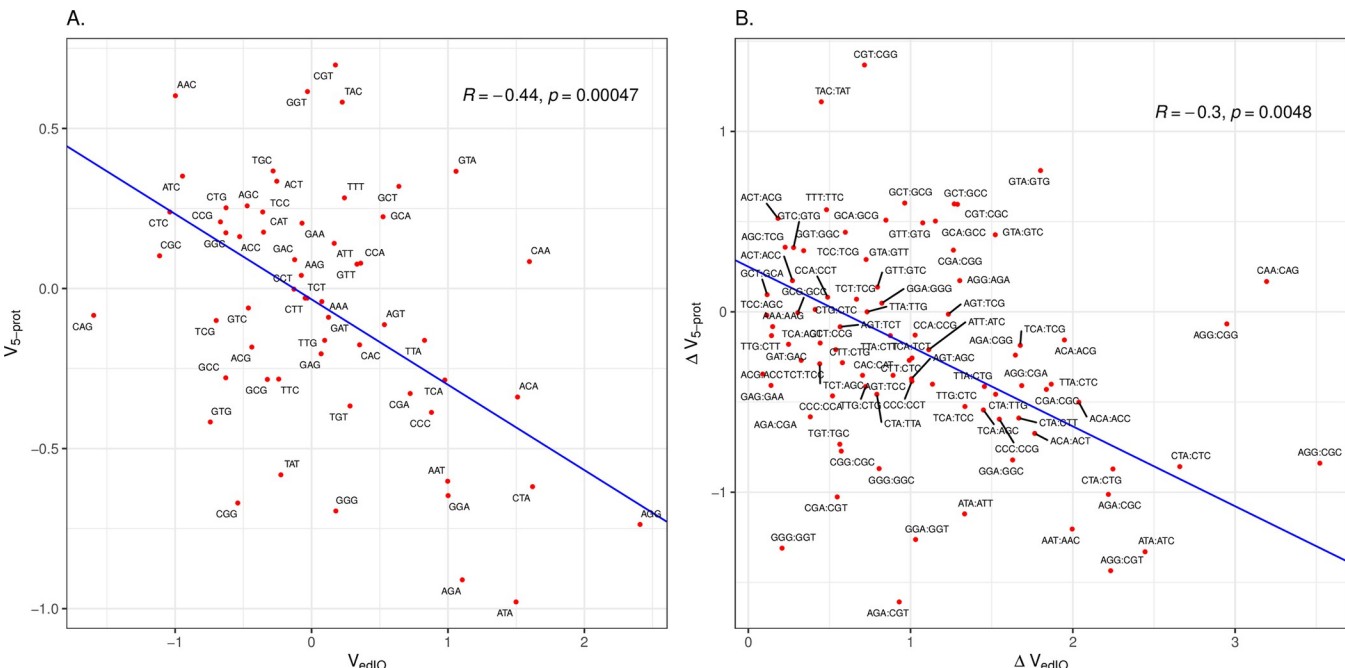

**Fig 2. Highly abundant genes in *E. coli* avoid codons associated with efficient translation initiation. A.** The *x* axis is the log odds ratio for the codon being associated with high protein expression levels when used at the 5'-end in experimental transgenes (edIO). The *y* axis is the log odds ratio for the codon being enriched at the 5'-ends of *E. coli* genes with high protein abundance compared to low protein abundance. Each data point is labelled as the codon it represents. **B.** As for Fig 2A, but comparing all pairwise combinations of synonymous codons (i.e. within the same codon block: N = 87). The pairwise differences are oriented such that, on the x axis, the codon with the lower value of the log odds ratio has its value subtracted from that of the higher value. The orientation is preserved for the y axis. This way no values on the x axis are negative. Each point is labelled by the oriented codon pair (first codon in the pair has the higher x-axis value, as seen in the Fig 2A). For both figures, Principle Components Analysis (PCA) was used to fit an orthogonal regression line. The Pearson's correlation coefficient and p-value are provided within the figures.

correlation, *rho* = -0.36, P = 0.0048; for Fig 2B: Spearman rank correlation, *rho* = -0.32, P = 0.0028). There are a few other genomes with good proteomic coverage and in none of these is their $V_{5\text{-prot}}$ vector positively correlated with the experimentally derived vector ($V_{edIO}$) (S1A–S1D Fig). In contrast to the experimental data [39], we see no relationship between protein level and predicted 5' end RNA stability (*rho* = 0.03, P = 0.055).

## Preference for translationally optimal and low noise associated codons explains why highly expressed genes do not employ codons that cause high expression

The above discovery prompts an obvious question: why do native highly expressed genes (defined at the protein level) over-employ codons that are, we presume, suppressive of initiation? Here we consider two possibilities centred on the hypothesis that the native HEGs are under conflicting or alternative selection pressures.

One such conflicting selection pressure could be greater selection to employ translationally optimal codons. To examine this model, we consider the relationship between the extent to which any codon is translational optimality and the codon usage at the 5' ends of the genes for the most common proteins (i.e. $V_{5\text{-prot}}$). We derive a comparable log odds ratio vector for translationally optimal codons by considering the core of native genes, comparing the 25% most highly expressed (defined as protein level) with the bottom 25% (N.B. as 3' ends may also be under selection for reduced stability [61], consideration of the core is better than consideration of the gene body as a whole). The resultant vector for translational optimality ($V_{TO}$)

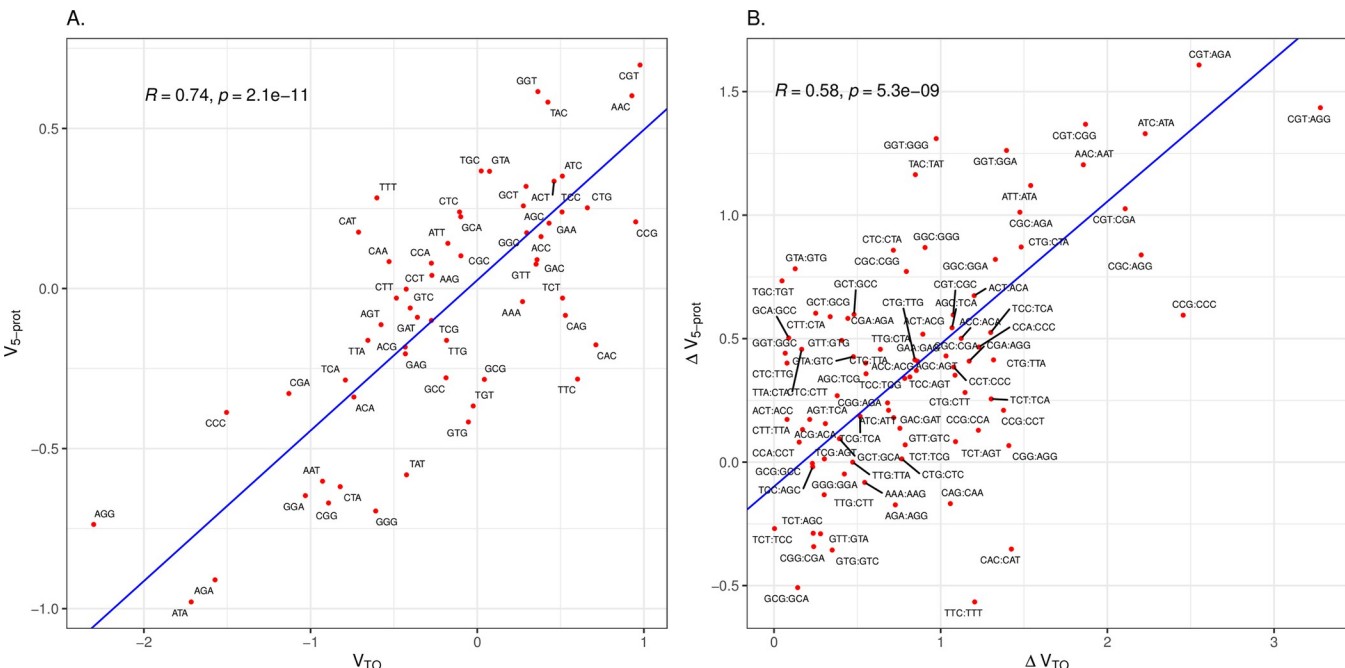

**Fig 3. The 5'-ends of *E. coli* genes with highly abundant proteins are enriched in codons associated with optimal translation. A.** The *x* axis is the log odds ratio for the codon being enriched in the cores of genes with high protein abundance compared to low protein abundance. The y axis is the log odds ratio for the codon being enriched at the 5'-ends of genes with high protein abundance compared to low protein abundance. Each data point is labelled as the codon it represents. **B.** As for Fig 3A, but comparing all pairwise combinations of synonymous codons (i.e. within the same codon block: N = 87). The pairwise differences are oriented such that, on the x axis, the codon with the lower value of the log odds ratio has its value subtracted from that of the higher value. The orientation is preserved for the y axis. This way no values on the x axis are negative. Each point is labelled by the oriented codon pair (first codon in the pair has the higher x-axis value, as seen in the Fig 3A). For both figures, Principle Components Analysis (PCA) was used to fit an orthogonal regression line. The Pearson's correlation coefficient and p-value are provided within the figures.

correlates well with the original w CAI scores from Sharp and Li [25](rho = 0.9, P<2 x 10$^{-16}$), as it should given that in principle they are measuring more or less the same thing (we presume that with better input data and exclusion of 5' and 3' effects $V_{TO}$ is a preferable measure for translational optimality than CAI). This metric of the degree of translational optimality also negatively correlates with that for $V_{edIO}$ (*r* = -0.64, P = 3.8 x 10$^{-8}$; S2A Fig) indicating that codons are typically either translational or initiation optimal. We identify only 7 of the 59 codons that have a positive log odds ratio on both scores (these being GTA, GCT, GTT, CAC, CGT, TAC, and AAA). The same negative correlation between translational optimality ($V_{TO}$) and initiation optimality ($V_{edIO}$) is seen when comparing in a pairwise manner all codons within a synonymous codon block (*r* = -0.44, P = 1.6 x 10$^{-5}$; S2B Fig), the codon with the higher initiation score tends to have a lower translational optimality score.

If then selection is conflicted at 5' ends in highly expressed native genes in *E. coli*, requiring both some level of translational optimality and initiation optimality, we might then predict that the codons enriched in the 5' ends in the genes for the most highly abundant proteins correspond to some degree with TO scores. Indeed, we find that *E. coli*'s V $_{5-prot}$ is positively correlated with $V_{TO}$ (Fig 3A and 3B). We conclude that HEGs may be under stronger selection to employ translationally optimal codons at their 5' ends (for caveat see Discussion).

Additionally, selection on native highly abundant proteins is not simply to mediate abundance but is also to enable low noise [80–82], noise being commonly defined as the ratio of the between-cell standard deviation in abundance divided by the between-cell mean abundance (i.e. the coefficient of variation). For native genes, those more dosage sensitive are expected to

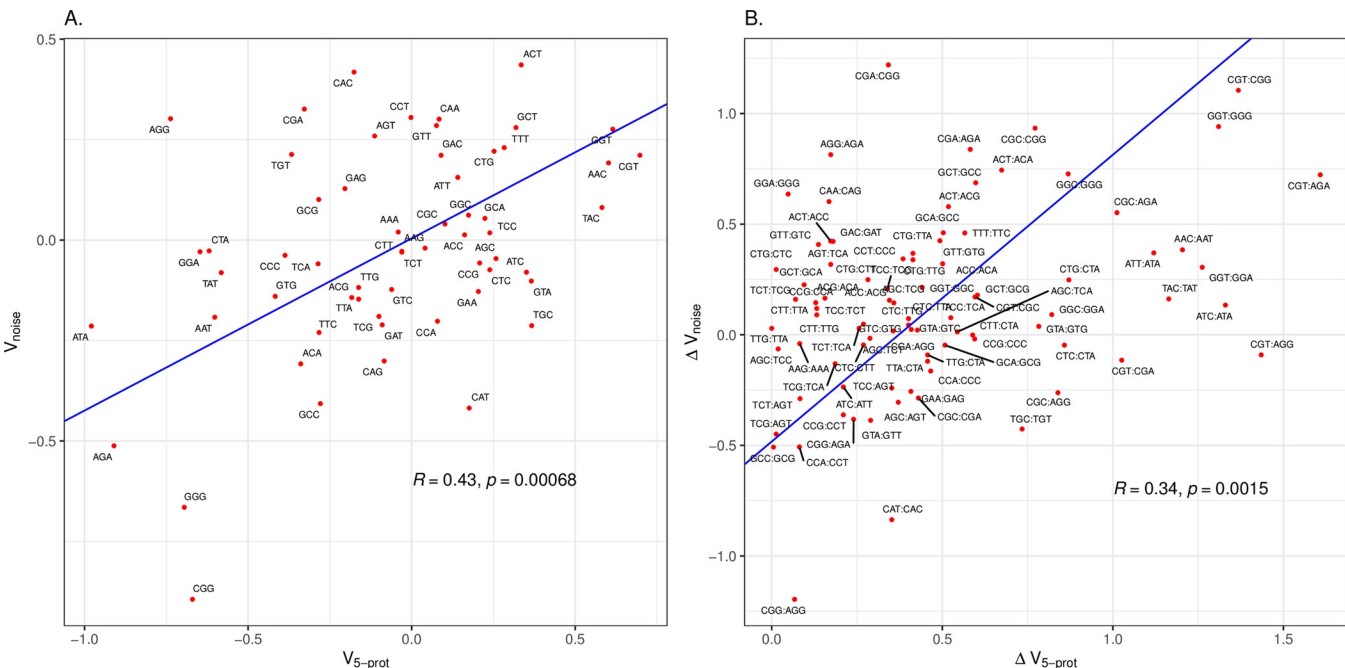

**Fig 4. The 5'-ends of *E. coli* genes with highly abundant proteins are enriched in codons associated with low noise. A.** The *x* axis is the log odds ratio for the codon being enriched at the 5'-ends of genes with high protein abundance compared to low protein abundance. The *y* axis is the log odds ratio for the codon being enriched at the 5'-ends of genes with low noise measurements compared to high noise. Each data point is labelled as the codon it represents. **B.** As for Fig 4A, but comparing all pairwise combinations of synonymous codons (i.e. within the same codon block: N = 87). The pairwise differences are oriented such that, on the x axis, the codon with the lower value of the log odds ratio has its value subtracted from that of the higher value. The orientation is preserved for the y axis. This way no values on the x axis are negative. Each point is labelled by the oriented codon pair (first codon in the pair has the higher x-axis value, as seen in the Fig 4A). For both figures, Principle Components Analysis (PCA) was used to fit an orthogonal regression line. The Pearson's correlation coefficient and p-value are provided within the figures.

be both more abundant and have lower noise, both devices to prevent dose stochastically reducing too far [80–82]. Consistent with reports that more ribosomes per transcript is associated with increased abundance-corrected noise [83] (but see also [84]), low noise is presumed to commonly be the result of having few ribosomes per transcript [85]. This in turn was evoked to explain why in bacteria many key genes have conserved inefficient ribosomal binding sites [83]. If true, then there could be selection on dose sensitive genes to have high transcript levels but low initiation rates. Such an effect would also be closely coupled with the possibility that low initiation also enables greater efficiency of ribosomal usage [86]. Both noise and efficiency may then select for low initiation rates for key genes. To examine this, we consider the correlation between 5' codons associated with low noise (directly measured) and codon preferences for the native highly abundant proteins. In addition, essential genes are, by definition, the most dose sensitive of genes as, when dose is zero fitness is likewise zero [80–82]. We thus also expect essential genes to be enriched at 5' ends (compared to non-essential genes) for codons associated with weak initiation.

From a data set of noise levels of native genes [87], we determine that 5' codons associated with low noise ($V_{noise}$). We observe that indeed, $V_{noise}$ is positively correlated with $V_{5\text{-prot}}$ (Fig 4A and 4B), indicating that the native genes with high protein levels prefer codons associated with low noise. This may, however, simply be because low noise genes are also the highly expressed genes. However, a multivariate model of $V_{5\text{-prot}}$ predicted by $V_{edIO}$, $V_{TO}$ and $V_{noise}$ reports independent effects of two parameters ($V_{TO}$, estimate = 0.34, P<0.0001, estimate = 0.41, $V_{noise}$, P<0.006) but not $V_{edIO}$.

Further, as expected if noise control is important, codons enriched 5' in essential genes compared to non-essential genes ($V_{ess}$) are enriched in 5' ends of the native highly expressed genes ($V_{ess}$~$V_{5\text{-}prot}$, $r$ = 0.62, P = 1.9 x $10^{-7}$; S3A Fig) but anti-correlated with the experimentally derived vector ($V_{ess}$~$V_{edIO}$, $r$ = -0.31, P = 0.018; S3B Fig). Both trends are seen when comparing within synonymous blocks but the latter is not significant (S3C and S3D Fig). These results again may be explained by essential genes having higher expression levels and thus enriched for translationally optimal codons. To address this, we consider how $V_{5\text{-}prot}$ is predicted by $V_{ess}$ and $V_{TO}$ in a multivariate regression. We find significant relationships for both predictors ($V_{TO}$ estimate = 0.33; $V_{ess}$ estimate = 0.42; p < 0.0001 for both).

### 5' ends of genes of *E. coli* and most other bacteria are enriched, compared to gene cores, for initiation optimal codons

The above results indicate that most probably owing to conflicting selection pressures, the most highly expressed native genes in *E. coli* avoid the initiation optimal codons. This result is not only surprising (given the classical mode to determine optimal codon usage) but also leaves a problem for transgene design: if we cannot define the initiation optimal set of 5' codons by the classical method of comparing highly and lowly expressed genes, how might we do it?

Classically, as tRNA profiles differ between species [88], we would not presume that translational optimality codon scores seen in one species need apply in any other [64]. However, if the mechanism of selection on 5' CDS codons is predominantly mediated by RNA stability/AU content, then the optimal initiation codons could also be initiation optimal in other bacteria [54] as the determinants of RNA stability should not, for the most part, be species-specific. Could we then just assume that the initiation optimality scores derived from the Gold standard *E. coli* transgene experiment [39] could be employed for transgenesis in other species?

How might we know if this would work? Naturally the ideal would be to repeat the massive transgene experiment for each target species. This we do not attempt. Rather, despite conflicting selection pressures on native gene 5' ends, we suggest that if the Gold-standard metric applies then a) in *E. coli* the overall 5' codon usage compared to CDS core should show a correspondence with the Gold standard set and, b) that the same metric (5' v core) for other bacteria should be correlated to a similar degree with the Gold-standard metric, as the *E. coli* metric is correlated with the Gold standard.

We start by asking whether, considering all genes, *E. coli* uses codons that are initiation optimal at the 5' ends, by considering the degree to which 5' ends of CDS are enriched/depleted for all codons compared to equally sized CDS core. We find that this 59 element vector ($V_{5\text{-}core}$) correlates well with that derived from the transgene experiment (r = 0.74, P = 2 x $10^{-11}$; Fig 5A). Comparing synonymous codons within a synonymous codon block reveals the same trend ($r$ = 0.43, P = 3 x $10^{-5}$; Fig 5B). We conclude that the experimental data predicts trends in codon usage particular to native 5' ends. AGG/AGA appear to be outliers.

This result is necessary but not sufficient. The key question is whether across bacteria we see similar enrichment at 5' ends compared to core? To address this for each of a multiplicity of phylogenetically distinct bacteria we derive a 59 element 5'-core log odds ratio vector. We then consider the correlation between each such vector and the Gold standard vector. If the *E. coli* experimental data has no value we expect the distribution of correlations to be centred on zero and no more than expected at 5% significance level. We find instead that the distribution is highly right shifted (Fig 6 and S2 Table) with 74% significantly positive after Bonferroni P value correction. The peak of occurrence is slightly under the value seen within *E. coli* ($r$ = 0.74), strongly supporting the notion that the Gold-standard experimental data from *E. coli* has much broader relevance.

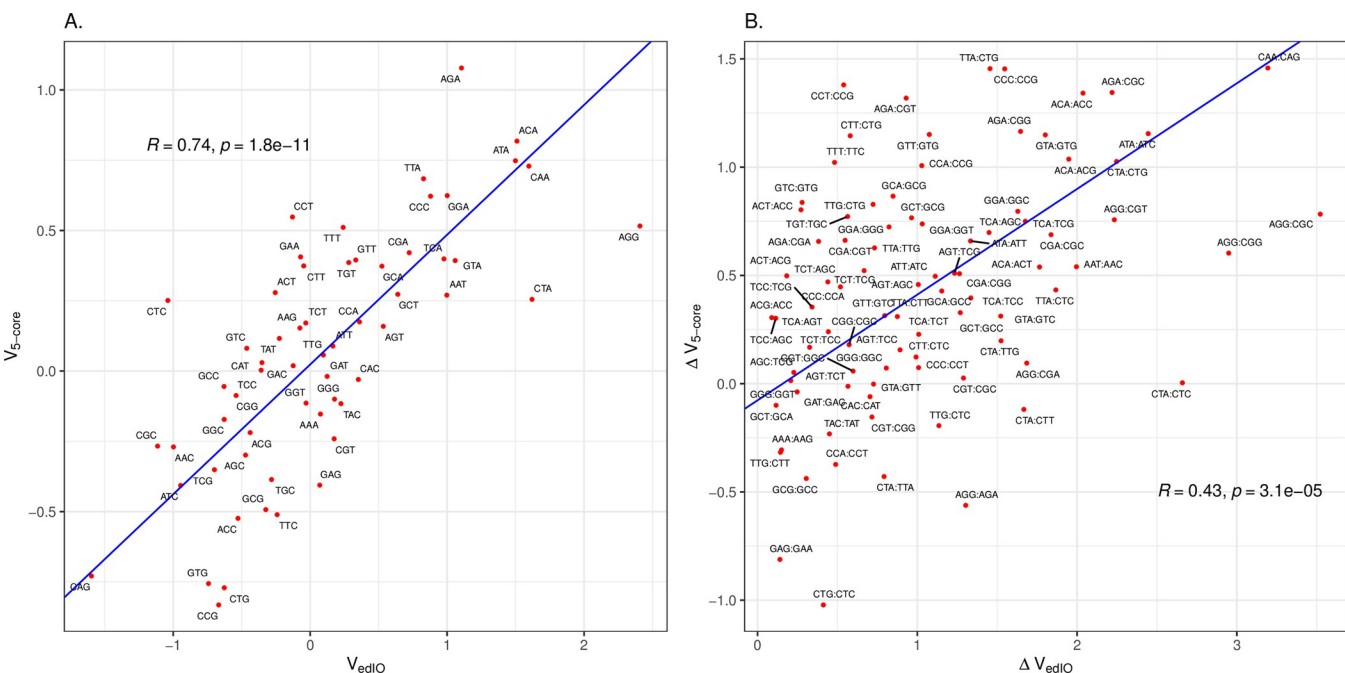

**Fig 5. The 5' ends of _E. coli_ genes are enriched in initiation optimal codons. A.** The _x_ axis is the log odds ratio for the codon being associated with high protein expression levels when used at the 5'-end in experimental transgenes (edIO). The _y_ axis is the log odds ratio for the codon being enriched at the 5'-ends of _E. coli_ genes compared to the gene cores. Each data point is labelled as the codon it represents. **B.** As for Fig 5A, but comparing all pairwise combinations of synonymous codons (i.e., within the same codon block: N = 87). The pairwise differences are oriented such that, on the x axis, the codon with the lower value of the log odds ratio has its value subtracted from that of the higher value. The orientation is preserved for the y axis. This way no values on the x axis are negative. Each point is labelled by the oriented codon pair (first codon in the pair has the higher x-axis value, as seen in the Fig 5A). For both figures, Principle Components Analysis (PCA) was used to fit an orthogonal regression line. The Pearson's correlation coefficient and p-value are provided within the figures.

We also wish to know the limits of possible applicability of the Gold standard data. We expect there to be exceptions for especially AT rich bacteria and those growing at high temperatures. We expect greater 5' enrichment of initiation optimal codons when GC pressure is high as here there will be stronger selection on the 5' ends for lower stability and, in turn, greater differentiation between 5' and core. By contrast, when genomic A/T content is high, low 5' mRNA stability can occur without selection and discrimination from the core codon content will be low. In addition, we expect a diminished signal when optimal growth temperature is high owing to weakened selection if 5' ends are thermally denaturing [51,89].

As expected, both GC pressure and thermal stability explain much of the variance in the correlations between each genomes' 5'-core enrichment trends and the Gold standard data. GC pressure, as measured by GC3 in CDS, is strongly predictive of the distribution of correlations (a quadratic fit is best with adjusted $r^2$ = 0.63, P = 2 x $10^{-16}$, Fig 7A). Higher growth temperature is associated with lower correlations, but the effect is weaker (adjusted $r^2$ = 0.1, P = 1 x $10^{-9}$). A joint model of the correlations predicted by GC3, GC3$^2$ and optimal growth temperature has a net adjusted $r^2$ of 0.65, P = 2 x $10^{-16}$ (GC3: estimate = 3.038, P < 2 x $10^{-16}$; GC3$^2$: estimate -2.27 P < 2 x 10–16, temp: estimate = -0.0027 P = 2.6 x $10^{-12}$; the interaction terms are all non-significant and so excluded). These two parameters thus can explain a considerable amount of the between-bacteria variation, with GC3 the more important of the two. While calculations of stability all come with serious caveats, as expected [54], 5' ends are less stable than core with the difference being larger when GC3 is larger ($r^2$ = 0.79, P< 2.2 x 10–16; S4 Fig).

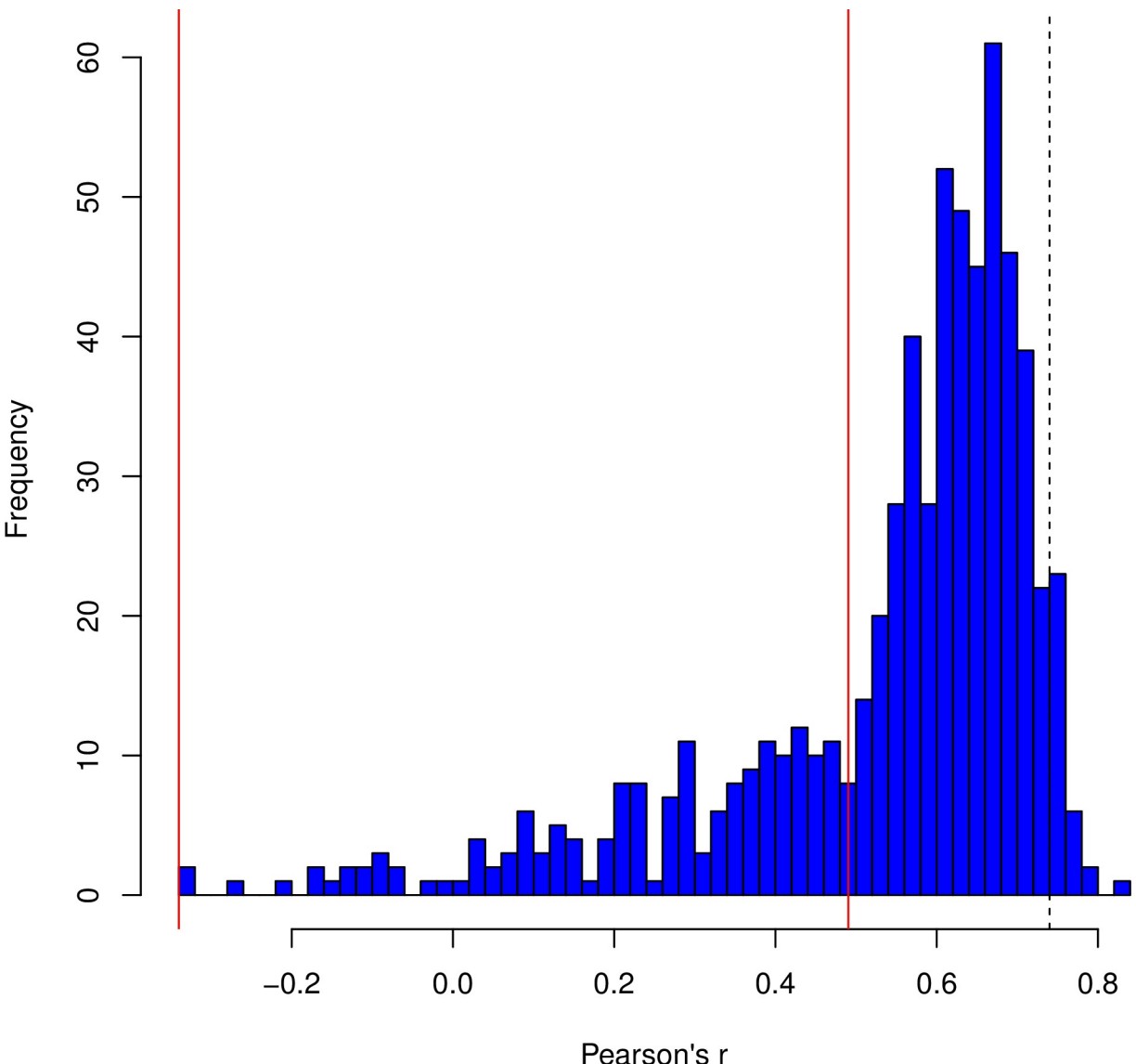

**Fig 6. Histogram of Pearson's R values from the correlation of $V_{edIO}$ with $V_{5'-core}$ from 650 bacterial genomes.** $V_{edIO}$ is the vector of log odds values for codon enrichment in the 5' ends of highly expressed experimental transgenes. $V_{5'-core}$ is the vector of log odds values for codon enrichment in the 5'-ends of genes compared to the core. The vertical solid red lines demarcate statistically significant correlations. P values are corrected for multiple testing using the Bonferroni correction. The vertical dashed black line indicates the value for the corresponding analysis within *E. coli*. The distribution is so right shifted away from the null of zero that no statistics are necessary.

## Discussion

Here we have determined the 5' codons whose presence is causative of increased protein levels in transgenes in *E. coli* and the trends for the 5' codons in *E. coli*'s natively highly expressed genes. Surprisingly, the 5' ends of the native genes for the most abundant proteins avoid the codons that are, we presume, initiation optimal. The likely reason for this is that in native highly expressed genes conflicting selecting pressures are acting. One explanation is that there is also selection for translationally optimal codons. For the most part the translationally optimal and initiation optimal codons do not greatly overlap, the former being GC rich [25,90], the latter AT rich [38,39]. We have highlighted the additional possible role of selection for

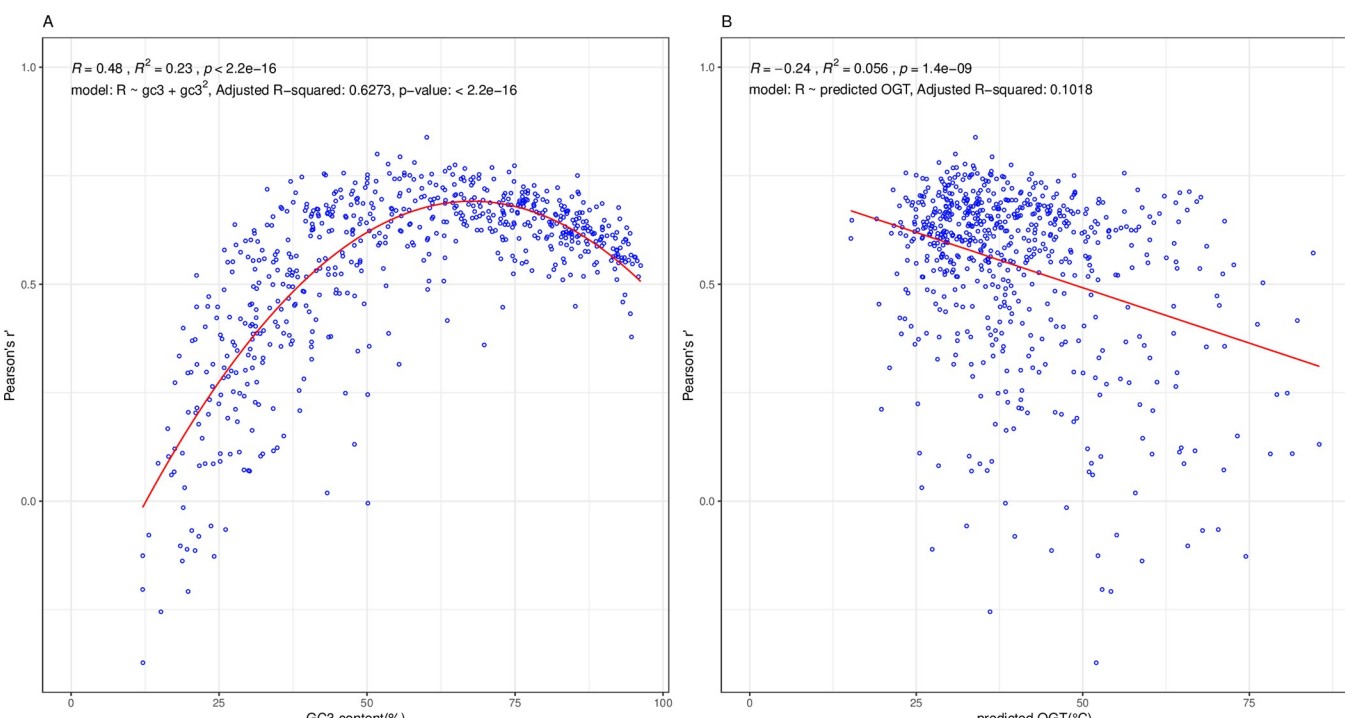

**Fig 7. Predictors of the correlation (y axis = Pearson's r) between codon enrichment at 5' ends ($V_{5\text{-core}}$) and the experimentally determined initiation codons ($V_{edIO}$) for 650 bacterial genomes. A.** Predicted by mean GC3 of all genes in a genome. The best fit is a quadratic line (shown). **B.** Predicted optimal growth temperature of the organism.

noise or efficiency both of which could be selection to decrease the initiation rate [85,86]. A combined statistical model suggests a role for both low initiation and translational optimality processes.

Whatever the cause of the incongruity of the 5' ends of highly expressed genes, the result questions the utility of employing highly expressed genes as the end point of a monotonic continuum in determining the direction of selection [68–76]. Our results then hold a cautionary tale as, were we to apply such a method and logic, the 5' codons we would derive are those causative of low, not high, protein abundance.

Furthermore, even within *E. coli*, we cannot go from the scores in the $V_{edIO}$ to derive the expected protein abundance of native genes, presumably because promoter, RNA half-life or protein half-life differences are key to explaining intragenomic variation in native protein levels. The 5' initiation optimal comparator metric to CAI for a non-model organism would envisage us going from $V_{5\text{-prot}}$, having defined a set of genes as highly expressed, to this CAI-like metric applicable to all genes. Were $V_{5\text{-prot}}$ and $V_{edIO}$ strongly positively correlated in *E. coli* this would have a robust defence. That simple defence is not there as the correlation is negative. Thus, unlike CAI which has the underpinning defence that overused codons in highly expressed genes specify more abundant iso-acceptor tRNAs, there is no easy mechanistic defence for extrapolating from 5' codon usage in native HEGs to protein abundance. Whether one would want to consider instead a machine learning approach without robust mechanistic defence is another issue.

The unexpected result also strongly argues against using the codon usage in highly expressed native genes as a guide to the codons to engineer into 5' ends of transgenes. Rather it seems to be more defendable simply to assume that the enrichment scores from the *E. coli*

transgene experiment might well have relevance in diverse bacteria. This makes sense in the context of prior data. As previously reported, these experimental data find that predominantly A (or T) ending codons promote high protein levels [39], this matching the overall preference for 5' codons to be A ending both in the model species (in which the transgene experiments have been performed) and more widely across bacteria [42]. This preference accords with the notion that, for most genes–just not highly expressed ones—the 5' ends are predominantly selected for low stability to enable ribosomal initiation [54]. We have also compared 5' and core of the genes of various bacterial genomes and indeed find that 5' ends tend to be low stability (at least as assayed by the Vienna package).

Given the correlations between the Gold standard experimental data and the 5'-core enrichments seen in many bacteria, utilization of the Gold standard set of codon enrichments from *E. coli*'s expression data is a defensible option to employ in other species to design 5' ends, especially for bacteria that have GC rich genomes and are not thermophiles. As the effect in GC poor genomes may be one of an inability to discriminate the initiation optimal codons when the gene body is AT rich, we do not presume that the Gold-standard metric would not be usefully applied in transgenes in these species, but rather that it is hard to know whether that would be true. The temperature effect is consistent with weaker selection on 5' stability at higher temperatures. If so, the Gold standard metric will be of less value but may nonetheless be a good starting point for transgene design.

Should we however just employ the Gold standard definition of initiation optimality or is there nothing to learn from the conflicting selection pressures affecting HEGs in *E. coli*? The role of noise in this context may, we suggest, be irrelevant. In transgenes we are not concerned that certain cells may produce lots of product and some little, while the same is not true as regards selection on native essential genes [80,81]. As such, for a transgene the mean (rather than the variance) output would appear to be the core concern. We could then engineer in more IO codons at a cost of increasing noise.

There may, by contrast, be a case to be made to consider the impact of translational selection or efficiency [86] in the design of 5' ends. We note, however, that as translationally optimal and initiation optimal codons tend to be anti-posed (S2 Fig) any selection to not employ initiation optimal codons (as with selection for low noise or high efficiency) will likely cause an increase in usage of translationally optimal codons, even if translational efficiency (speed or accuracy) is not a focus of selection. In this sense we cannot be certain that the increase in TO codons in the most abundant proteins is owing to selection for TO *per se* so much as selection against initiation optimality. Nonetheless, while TO codons do not greatly affect the net productivity of the transgenes, they do affect cell fitness [38,43], ribosomal hogging by non-optimal codons being a possible mechanism. Reduced efficiency [86] is likewise expected to infer costs. As such there may be an argument that net productivity, which may factor in the cell viability costs of bearing a highly expressed slowly translated mRNA, would require the replacement of IO codons with TO ones in the 5' end, just as native highly expressed proteins do. Usage of the 7 codons that are positive for IO and TO would be an obvious starting position (these being GTA, GCT, GTT, CAC, CGT, TAC, and AAA). More generally, the usage of log odds ratio vectors of (species specific) TO and of Gold standard IO scores make for potentially easy to generate candidate optimal 5' transgene vectors. One need merely specify the relative balance of IO and TO to derive by vector multiplication an input vector. The resulting vector can then be employed to stochastically determine codon usage for any given 5' end, preserving the amino acid usage. *A priori* the optimal weighting is not obvious and is best considered an empirical issue.

One limitation of our work is that we have not considered codon-by-position affects. Prior work has, for example, claimed that NGG codons at positions 2, 3 and 5 are suppressive [91].

This strongly contrasts with our result that AGG and GGG have a positive log odds ratios in $V_{edIO}$ and only CGG is negative (TGG being non-redundant has no score). Despite necessary caveats about low resolution of our data when split by position, the discrepancy appears not, however, to be owing to site-specificity. When we break down our log odds ratios by codon and by position for the experimental data (S3 Table), AGG has a positive log odds ratio at all positions, while CGG has a negative score at all but one position. GGG is more complex having 5 positive values and 5 negative values. It is however positive for all of the supposedly suppressive positions (2, 3 and 5). There is thus no evidence that the discrepancy is owing to position specific effects being overwhelmed by trends at the other positions. Indeed, more generally we see no evidence for position effects. One way to address this is to ask about the degree of concordance between the 10 different 59 element vectors from codon positions 2 to 11 in the experimental data. Applying the internal consistency test, Cronbach's alpha, we see "excellent" [92] concordance (alpha = 0.948, 95% CI 0.905–0.967) indicating no good evidence for any N terminal position being different from any other. The ten 59 element vectors for positions 2–11 also are very strongly correlated one with another (S4 Table). Rather, the most likely reason for the discrepancy is that the prior analysis [91] compared between constructs that differ in amino acid content, while our metric exclusively considers effects comparable to other codons specifying the same amino acid, this being of the greatest relevance in designing transgenes and understanding selection on synonymous sites. We can conclude that we see no *prima facie* support for the hypothesis that NGG codons are suppressive *relative to synonyms* (a claim not previously made).

Despite the above, the extremely high overall IO value of AGG (Fig 1) is an enigma worthy of further investigation. It isn't a sample size artefact, the standard error being relatively small (Fig 1). However, the value for any given codon is also dependent on the usage of the synonyms. In this regard, AGG is unusual in that it has four synonyms that are GC rich. AGGs high value may thus in part reflect a more general trend for high log odds ratios of the two AG starting codons for arginine, as opposed to the CG[G/C] ones (N.B. within the 4 fold block the *A* ending codon has the highest ratio). Indeed, if we define AGA and AGG as a two-fold block then the log odds ratio for AGG enrichment is 1.12, which is no longer the highest value. Nonetheless, there remains the mystery as to why AGG has a higher logs odds ratio than AGA, the codon that more obviously obeys the A/T over G/C preference. One possible clue comes from comparison of $V_{TO}$ and $V_{edIO}$ scores. It can be no accident, we suggest, that AGG both has the highest log odds ratio for the later and the lowest for the former (S2 Fig). This would support the hypothesis that selection on 5' ends may not simply be mediated by selection for low stability but also for translational non-optimality, possibly owing to effects on ribosomal velocity [61] (but see [55]). Similarly, while within the 6-fold leucine block the two A ending codons (CTA, TTA) have the highest log odds in the Goodman et al data [39], *prima facie* CTA should have the lower value given its first site usage but doesn't. However, consistent with a possible effect of ribosomal speed, CTA has a lower TO score than TTA.

Perhaps the most important caveat to our analysis concerns the assumption that the Goodman et al. data provide a Gold standard for estimation of initiation optimality scores (initiation optimality being broadly defined). The values we have derived will no doubt vary with the thresholds for the most abundant and least abundant classes (we employ top and bottom 25% by Prot.FCC). Perhaps more importantly, our metric does not factor in any aspects of context. This could mean codon-by-position effects (above) or, possibly more importantly, the impact of the 5' UTR employed. Indeed, if the mechanism of action is mediated by RNA folding and stability effects, we might expect that the 5' UTR composition would be important in determining the initiation optimality scores for the 5' codons. To determine this one would need a

more extensive experiment in which amino acids, codons and 5'UTR are all extensively randomised. We are unaware of any such experiment.

While in principle a serious issue, we have reason to suppose that our IO scores are, however, fairly robust. First, we recover a trend for high A/U content promoting expression as do all prior experiments [38,41–43]. Second, this A/U preference is seen in cross-species analyses [42]. Third, the log odds enrichment scores appear to be highly consistent between all N terminal positions (see above). Were context to be especially influential we would have expected that log odds ratios would show more position context dependency. Fourth, in *E. coli* there is a robust correlation ($r = 0.74$) between the Gold standard vector and 5' enrichment vector (Fig 5). As this is in part diminished by inclusion of the highly expressed genes, we can also see what happens when we break the data in to bins by expression level, derive a 5' enrichment vector for each bin and consider how these vectors then correlate with our Gold standard vector. Doing this we find that the correlation can exceed $r = 0.8$, the correlation being weakest for both the most highly expressed (under conflicting selection) and the lowest expressed (probably under the weakest selection) native genes (S5 Fig). That the Gold standard vector also predicts 5' enrichment in many bacteria (Fig 6) similarly suggests that it isn't too misleading. We consider it viable to still consider it the present Gold standard (best current estimate), but whether it is definitive requires further experimentation. We suggest that it is robust enough both to substantiate our claim that HEGs avoid IO codons when enrichment is defined with respect to LEGs (Fig 2) and of utility for the design of 5' ends of transgenes. In addition, the 5'-core enrichment trends also seem indicative of IO codon status especially for genes that are neither lowly or highly expressed (S5 Fig) suggesting a further potential metric for transgene design, potentially especially relevant for GC rich bacteria.

A further limitation is that we have restricted analysis to bacteria. While a trend for low RNA stability at CDS 5' ends is thought to be common [54], at least in mammals the reverse nucleotide trend is observed, i.e. high GC at CDS 5' ends [93], this matching transgene analysis reporting that high GC, not low GC as in bacteria, is associated with high protein levels [93–95]. This effect is mediated in part by nuclear export favouring GC rich transcripts [93,95], a feature of no relevance to bacteria. The limitations of the extension of the *E. coli* data outside of bacteria we leave to future study.

## Materials and methods

### Data analysis

Data analysis was carried out using Python version 3.11 with custom written scripts. Plots were generated in R version 4.1.3 [96] using Base-R and the package ggplot2 [97]. All relevant scrips (detailing their package requirements) and their output files are available from 10.5281/zenodo.8349580.

### Genome sequence sources and genome parsing

For *Escherichia coli*, the full genome assembly for K-12 strain MG1655 [accession: NC_000913] was downloaded in GenBank format from NCBI in March 2023. The coding sequences (CDS) of all genes were extracted using the SeqIO package from Biopython [98].

For the analysis of other bacteria the genome annotations were downloaded from EMBL (http://www.ebi.ac.uk/genomes/bacteria.html). These were filtered so that all genomes were a minimum of 500,000 bp in length, ensuring any within-genome calculations had an adequate sample size. The genomes selected all belonged to different genera (i.e. only one genome per Genus), this being done to minimise the chance of skewed results due to phylogenetic inertia. See 10.5281/zenodo.8349580 for the full list of included bacteria and their accession numbers.

For each genome, a custom Python script was used to extract each CDS and output them in FASTA format. Sequences that weren't divisible by 3, didn't end with a stop codon, contained an internal stop codon or didn't begin with a start codon (NTG) were removed.

## Synonymous codon enrichment calculations

Throughout the study, log odds ratios [99] were used to consider the enrichment of each codon in specific target sequences relative to its synonyms. The general equation used to calculate the relative enrichment of a codon in a target set of sequences compared to a second set of sequences is given as:

$$\text{odds ratio} = \frac{N_{codon}1}{N_{synonymous}1} \div \frac{N_{codon}2}{N_{synonymous}2}$$

$$\text{logodds ratio} = \text{Ln}(oddsratio)$$

where $N_{codon}1$ is the count of the focal codon in the target set of sequences, $N_{synonymous}1$ is the count of its synonyms in the target set of sequences, $N_{codon}2$ is the count of the focal codon in the second set of sequences and $N_{synonymous}2$ is the count of its synonyms in the second set of sequences. We then derive a 59 element vector of log odds ratios, one value for each codon with at least one synonym, this being independent of amino acid usage. Standard error of the log odds ratio is the square root of the sum of the inverse of the above four values [99] i.e. the root of $1/N_{codon}1 + 1/N_{codon}2 + 1/N_{synonymous}1 + 1/N_{synonymous}2$. This was then employed in the following manners:

**Codons enriched at the 5'-ends of highly expressed E. coli transgenes.** To assess the experimentally derived initiation degree of optimality (edIO) of each codon, a log odds ratio was calculated describing its enrichment at the 5'-ends of highly expressed experimental transgenes compared to lowly expressed transgenes, allowing for variation in transcript abundance. The data comes from a large-scale transgene experiment in *E. coli* that measured the expression levels of 14,234 constructs with different combinations of promoters, ribosomal binding site (RBS) and N-terminal sequence [39] fused to an sfGFP reporter, the level of protein derived being compared to an mCherry reporter under a constitutive promoter on the same plasmid. The amino acid content of the N terminal ends (first 11 codons) was taken from 137 different native *E. coli* essential genes. Because of the employment of multiple 5' ends differing not only in codon usage but also amino content, this data set is unique in that we can obtain log odds ratios for all 59 codons that have at least one synonym. The same is not true for any alternative data set as they preserve the 5' amino acid content of the construct.

For each N terminus, promotor, RBS combination Goodman et al [39] generated 13 different constructs differing only in synonymous site content in the first 11 codons. One of the 13 was designed to employ translationally optimal codons, one translationally non-optimal codons and the others varied in predicted RNA stability. For each group of 13 they determined fold enrichment of all constructs (compared with the mCherry control) against the mean of the 13, this being their metric Prot.FCC. This is comparable between groups of 13 [39]. Note that if there were amino acid level effects on protein abundance, such as an effect of positively charged amino acids at N termini [61] (but see [100]), this effect is irrelevant in the current context as we only compare between constructs with the same N terminal region.

Here, the target set of sequences was the 5'-ends of the top 25% of constructs by Prot.FCC (normalised protein level), and the second set of sequences was the 5'-ends of the bottom 25% of constructs by Prot.FCC. All constructs were considered even if their precise Prot.FCC level was out of range (for the highest and lowest we need only consider that they were extreme, not

how extreme). The 59 values constitute the experimentally determined vector of IO values, $V_{edIO}$.

**Codons enriched at the 5'-ends of bacterial genes compared to the core.** To look at general trends of 5' codon usage across bacterial genes, log odds ratios were calculated for the enrichment of each codon at 5'-ends compared to CDS cores in any given genome. Here, the target set of sequences was the 5'-ends of the CDSs, and the second set of sequences was the cores of the CDSs for a given bacteria. Throughout this study, the 5'-end is defined as the first 11 codons of the ORF, excluding the start codon. This region was selected as previous experiments have reported that the first 30–40 nucleotides of the CDS significantly impact translation initiation, while any changes in codon usage further downstream have minimal influence on expression [38,39]. An equal length sequence was taken from the middle of the CDS to represent the gene core. This was preferable to looking at the whole gene body as 3'-ends may also be under selection for lower stability (Tuller et al., 2011), confounding comparisons with 5'-ends.

**Codons enriched at the 5'-ends of E. coli genes with high protein abundance.** To assess codon usage at the 5'-ends of highly expressed genes, log odds ratios were calculated for the enrichment of each codon at 5'-ends of genes with high protein abundance compared to low protein abundance. Absolute protein abundance data was downloaded from PaxDb [79] on 07/03/23 for the five species of bacterium that had integrated data sets with over 70% coverage (*Escherichia coli*– 99% coverage, *Bacillus subtilis*– 96%, *Mycobacterium tuberculosis*– 85%, *Microcystic aeruginosa*– 79% *and Pseudomonas aeruginosa*– 74%). For each bacterium, the target set of sequences was the 5'-ends of the top 25% of genes by absolute protein abundance and the second set of sequences was the 5'-ends of the bottom 25% of genes by absolute protein abundance.

**Codons enriched in the cores of E. coli genes with high protein abundance.** To assess codon usage in the core of highly expressed *E. coli* genes, log odds ratios were calculated for the enrichment of each codon in the cores of genes with high protein abundance compared to low protein abundance, using the same PaxDb data. Here, the target set of sequences was the cores of the top 25% of genes by absolute protein abundance and the second set of sequences was the cores of the bottom 25% of genes by absolute protein abundance. This metric we assume to a measure of translational optimality and indeed correlates robustly with the original Sharp and Li [25] $w$ metric (rho = 0.9, P $<$ 2 x $10^{-16}$).

**Codons enriched at the 5'-ends of E. coli genes with low noise.** To explore the relationship between 5' CDS codon usage in *E. coli* and noise, log odds ratios were calculated for the enrichment of each codon at the 5'-ends of genes with low noise compared to high noise. Direct measurements of noise in protein expression were retrieved from [87]. Here, the target set of sequences was the 5'-ends of the bottom 40% of genes by Noise_Protein and the second set of sequences was the 5'-ends of the top 40% of genes by Noise_Protein. Top and bottom 40% were used to maximise the sample sizes for the calculation, as this data set only contained ~1000 genes.

**Codons enriched at the 5'-ends of essential E. coli genes.** To look at 5' codon usage in *E. coli* essential genes, log odds ratios were calculated for the enrichment of each codon at the 5'-ends of essential genes compared to non-essential genes. The list of predicted essential genes based on sequencing of a high-density transposon library was obtained from [101]. Here, the target set of sequences was the 5'-ends of all predicted essential genes and the second set of sequences was the 5'-ends of all nonessential genes.

For all log odds ratio vectors for *E.coli* see S1 Table. For all log odds ratio vectors for *5' core* comparisons for all other bacteria see S2 Table. For log odds ratio vectors for *E.coli* from experimental transgene data separated by codon position, see S3 Table.

## Comparison between vectors

We employ two methods to compare any two vectors of log odds ratios. In the first we consider a Pearson product moment correlation between the two vectors (to check for outlier effects we also employ Spearman's rank correlation where stated). This may however have a statistical problem given the nature of the log odds ratio calculation, namely that the values for any codon are not independent of the values for the synonyms. Indeed, for a two-fold codon set if the log odds ratio of one is $x$, then the other must be exactly -$x$. As with phylogenetic non-independence, there may be a problem of inflation of sample size by considering both $x$ and -$x$. To be cautious that results are not artefacts of this non-independence, following [102] we consider the difference between log odds ratios between all pairs of synonymous codons, each difference being a data point. In the case of two-fold codons for example, we employ one difference value as one data point rather than two ($x$ and -$x$). For each comparison we present the correlation of the raw values (as plot A) and the correlation of the pairwise difference plot (as plot B).

## Estimation of optimal growth temperature

To estimate optimal growth temperatures, we employed CnnPOGTP, a k-mer based machine learning approach to predict OGTs [103]. The genomic sequence of each of our bacteria was provided (in fasta format) to CnnPOGTP [103] using a python script with the selenium webdriver package directed to its website (http://www.orgene.net/CnnPOGTP/). These estimates we refer to as predicted Optimal Growth Temperatures (pOGT).

To validate the predictions, we considered the relationship between these predictions and experimentally determined values from databases TEMPURA [104], ThermoBase [105] and GOSHA [106]. Only the results that had "Complete" level genomic information were taken from the GOSHA database. Of 650 bacterial genomes, 335 had a measured OGT. The correlation between the pOGTs and the measured OGTs for these 335 had an $r^2$ value of 0.905. Given this high level of concordance, and to enable as large a dataset as possible, for all analyses we employed the pOGT values.

## Estimation of RNA stability

To obtain the mean stabilities of the 5' and core sites for each bacterium, the sequences of the first and middle 12 codons of each gene were extracted. These sequences were then provided to the Vienna RNAfold server [107] (http://rna.tbi.univie.ac.at/cgi-bin/RNAWebSuite/RNAfold.cgi) via python script using the selenium webdriver package, with the estimated Gibbs free energy of each gene then being extracted. The mean of these were then taken for all the 5' and core sequences for each genome.

## Supporting information

**S1 Table. Log odds ratios for each codon with at least one synonym for the 6 E. coli specific features.** VedIO: 5' enrichment in the experimental transgene data (top 25% by Prot. FCC v bottom 25%); V5-core: enrichment in the 5' end of all E. coli native genes versus the core. V5-prot: 5' enrichment in the native gene data (top 25% by protein abundance v bottom 25%); VTO: core CDS enrichment in the native gene data (top 25% by protein abundance v bottom 25%); Vnoise: 5' enrichment in the low noise native gene data (bottom 25% by noise v top 25%); Vess: 5' enrichment in essential genes (essential v non-essentials).
(CSV)

**S2 Table. Log odds ratios for 5' v core enrichment for multiple bacterial genomes (indicated by accession number).**
(CSV)

**S3 Table. Log odds ratio of 5' enrichment in the E. coli experimental transgene data (top 25% by Prot. FCC v bottom 25%) separated by codon position.**
(CSV)

**S4 Table. Pearson product moment correlations between the 10 59 element vectors of log odds for all codons at N terminal amino acid positions 2–11 in the Goodman et al [39] data.** Input for this calculation was S3 Table omitting the rows for stop codons. Stars indicate degree of significance (* = 0.05, ** = 0.01, *** = 0.001, **** = 0.0001).
(PDF)

**S1 Fig. $V_{edIO}$ does not predict patterns of codon usage at the 5' ends of genes with highly abundant proteins in four different bacterial genomes.**
(PDF)

**S2 Fig. The relationship between translational optimality scores and initiation optimal scores.**
(PDF)

**S3 Fig. Codon usage in Essential genes.**
(PDF)

**S4 Fig. The relationship between GC3 and the difference in predicted stability of the 5' ends and core sections of all genes in each of 650 genomes.**
(PDF)

**S5 Fig. The relationship between the 5' enrichment vector and $V_{edIO}$ for bins ordered by expression level.** The E. coli genes are rank ordered by their protein abundance (expression level). They are then split into overlapping bins of various sizes (1/4 the total, $1/5^{th}$ the total etc). For each bin of genes a 5' enrichment score (compared to core) is calculated for each codon. This vector is then compared with $V_{edIO}$ the y axis being the correlation between the two. The horizontal dashed line is the comparable correlation considering all genes. The bins are ordered left (lowest expression) to right (highest expression bin). The x axis value is the rank order percentile position of the lowest expressed gene in the given bin. Note that correlation is lowest for both the most highly expressed and most lowly expressed genes. Intermediate bins give correlations in excess of r = 0.8.
(PDF)

## Author Contributions

**Conceptualization:** Laurence D. Hurst.

**Data curation:** Loveday E. Lewin, Kate G. Daniels, Laurence D. Hurst.

**Formal analysis:** Loveday E. Lewin, Kate G. Daniels, Laurence D. Hurst.

**Investigation:** Loveday E. Lewin, Kate G. Daniels, Laurence D. Hurst.

**Methodology:** Loveday E. Lewin, Kate G. Daniels, Laurence D. Hurst.

**Project administration:** Laurence D. Hurst.

**Resources:** Loveday E. Lewin, Kate G. Daniels, Laurence D. Hurst.

**Software:** Loveday E. Lewin, Kate G. Daniels, Laurence D. Hurst.

**Supervision:** Laurence D. Hurst.

**Validation:** Loveday E. Lewin, Kate G. Daniels, Laurence D. Hurst.

**Visualization:** Loveday E. Lewin, Kate G. Daniels, Laurence D. Hurst.

**Writing – original draft:** Laurence D. Hurst.

**Writing – review & editing:** Loveday E. Lewin, Laurence D. Hurst.

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
