## [Decision Letter · Decision Letter 0]

6 Sep 2023

Dear Prof Hurst,

Thank you very much for submitting your manuscript "Genes for highly abundant proteins in Escherichia coli avoid 5’ codons that promote ribosomal initiation." for consideration at PLOS Computational Biology.

As with all papers reviewed by the journal, your manuscript was reviewed by members of the editorial board and by several independent reviewers. In light of the reviews (below this email), we would like to invite the resubmission of a significantly-revised version that takes into account the reviewers' comments.

We cannot make any decision about publication until we have seen the revised manuscript and your response to the reviewers' comments. Your revised manuscript is also likely to be sent to reviewers for further evaluation.

Sincerely,

Marc Robinson-Rechavi

Academic Editor

PLOS Computational Biology

Pedro Mendes

Section Editor

PLOS Computational Biology

Reviewer's Responses to Questions

**Comments to the Authors:**

Reviewer #1: In this manuscript, Lewin et al. computationally analyze the codon usage within the 5’ region of bacterial mRNAs from indicated categories (sets). For this purpose, the authors define the “log odds ratio” to quantify the relative enrichment of a codon between two sets of mRNAs to be compared. Some interesting features on the 5’ region of mRNA have been concluded from these analyses, such as codons ending with A or U promoting high protein levels, enrichment of optimal codons for translational initiation. The most intriguing finding is that, in E. coli, genes for highly abundant protein avoid 5’ codons that promote translational initiation, as illustrated by the title. Overall, the results from this manuscript provide insights into the mechanism of translation in bacteria and will be useful for the design of expression vectors. The followings are some concerns I have.

In Fig. 1, the AGG codon (for arginine) shows the highest log odds ratio among all 59 codons, especially it is an exception (ending with non-A/U). This phenomenon deserves more discussion. In particular, the AGG codon is a rare codon and, as the authors mentioned (reference #85), NGG (including AGG) codons are suppressive in the 5’ region of mRNA. Also, the y axis label of this figure should be “log odds ratio”.

Other minor points:

* RNA “stability” used in this manuscript is exclusively to indicate the “structural stability” of RNA. However, some readers of different backgrounds may mis-regard it as, say, the resistance of RNA to degradation. The authors should make it clear in the context.

* p.15, line 2 from bottom: My understanding is that the histogram per se in Fig. 6 is “left skewed”, though the authors would emphasize that the distribution is largely shifted to the right from zero.

* p.16, line 8: “discrimination from the core codon content will we low.” A typo: we -> be

* p.19, line 10 from bottom: “Usage of the 7 codons that are positive for IO and TO would be an obvious starting position.” What are the 7 codons?

Reviewer #2: Lewin and colleagues perform an in silico analysis of nucleotide sequences in general and codon sequences in particular. Independent recent studies concluded that synonymous codon modifications that reduce 5’ mRNA stability result in increased protein levels. The goal of the present study is to reveal relevant details that determine the expression levels in E.coli, both of natural genes and of transgenes. Surprisingly, this study revealed that the 5’ ends of native genes that specify highly abundant proteins avoid experimentally demonstrable initiation optimal codons. An interesting study, but there are concerns that should be addressed.

Major comments:

1. Although the method employed in this study appears valid and yields interesting data regarding the codon usage in HEGs in E. coli there are some conceptual unclarities in this work that require further elaboration. As the authors describe in the introduction, various studies (Kudla et al., Goodman et al., Nieuwkoop et al., Hollerer & Jeschek) convincingly demonstrate that codon frequency and/or tRNA abundance do not directly reflect the corresponding protein level, but rather that this appears to be better predicted by the potential mRNA folding of the region around the translation start at the mRNA transcript, including the 5’UTR and the 5’CDS. Therefore, one could conclude that the ‘optimal initiation codons’ would be context-dependent, and are dependent the sequence of the 5’CDS and 5’UTR. With this in mind, could the authors explain why they still decided to use codon usage as a sole measure for translation efficiency? Would the RNA stability not be a better indicator for expression level? The Golden standard dataset reports data on 5’ RNA stability for the analyzed constructs. Could a similar analysis be done for the selected HEG dataset of this study and could information on the folding energies of the studied HEGs reveal additional insight in the ‘choice’ of codon usage of HEGs?

2. An important question is whether the Golden standard dataset the right reference for this approach? Since the Golden standard dataset is a result of a library where 5’UTRs and 5’CDSs were (randomly) combined, is it not possible that Goodman et al. selected for combinations of 5’UTRs and 5’CDSs with A/T-rich, rare codons resulting in e.g. low folding energies? This is very useful information for transgene expression, but do the sequences in the library of Goodman et al. also reflect the in vivo conditions of HEGs in E. coli and does that allow for direct comparison of the correlation between expression level and codon usage? Do the expression levels achieved by Goodman et al. match the expression levels of the studied HEGs, or are they higher/lower? Goodman et al. used two different promoters which allows for normalization on transcript level between 2 datasets, while HEGs will each be under control of their own promoters. Could there also be an additional effect of transcript level on the expression level of the HEGs? Please elaborate. Also note that the dataset used by Goodman et al. did not employ 147 different target genes (as the authors mention in the Materials and Methods (page 22, first paragraph)), but the 11 first amino acids of 137 endogenous essential E. coli genes cloned at the 5’end of an sfGFP reporter gene. The statement that this dataset is superior as it employed 147 target genes rather than just one target gene in other studies therefore is misleading and should be rephrased.

3. The introduction is rather lengthy and contains a lot of ‘woolly’ language, convoluted sentences and repetition. Overall, this makes it difficult to read. Please make the introduction more concise. Specifically, pages 3 and 4 all describe the single concept of codon usage not relating to protein expression level which could be significantly condensed. Also, the rationale of the study (pages 6 and 7) repeats many concepts previously introduced.

4. In the introduction (page 5, last paragraph), the authors provide a description of IO and TO codons, essential for further understanding of the paper, but the given definition or distinction between the two is fuzzy and sometimes contradicting. What exactly constitutes either a TO or IO codon, is there a threshold or definition? Could A/T-rich TO codons also be IO codons? Please clarify a (concrete) distinction between the two.

5. In the Results (figures 2 – 5, and page 9), the rationale behind the data representation in panel B is not entirely clear. It is clear how the data in the figure is obtained, but it is not explained what new information this representation of data brings, especially as it not only compares the expression between IO and TO-codons, but also includes comparison within ‘non-IO’ vs. ‘non-IO’ and ‘non-TO’ vs ‘non-TO’ codons with each other. Does this not introduce a lot of noise? Please clarify the rationale behind this data representation in more detail.

Minor comments:

6. Page 2 - line 4: “concept of the translationally “optimal” codon”: The definition is not clear, is this the most abundant codon or the codons present in HEGs? The next sentence suggests this is not the same.

7. Page 4 - Paragraph 2: “While this accords with […] processing be too slow” This long sentence is very convoluted and it is unclear what the authors mean, please rephrase.

8. Page 4 - last paragraph, first sentence: Misses a word (assuming the effect of?)

9. Page 5 - paragraph 1: Typos: Hypothesied (Hypothesized), Nieukoop (Nieuwkoop)

10. Page 5 - paragraph 2: The related ramp hypothesis is not only dependent on 5’ folding energy, but the cited paper shows it is a combination of folding energy, charge and codon bias of the sequence, indicating there is a (partial) CAI effect on the 5’CDS. How would this relate to the here presented findings?

11. Page 6 - last paragraph: The authors state that the presented method “permits us to understand the general utility of employing highly expressed genes as the end point of a monotonic continuum in determining the direction of selection.” The description is vague, how do the given examples of this relate to optimality of initiation codons?

12. Results, first paragraph: The study describes an analysis performed on the ‘5’ end of mRNA’ of the Goodman dataset and HEGs. Could the authors specify (here, and in M&M) which cut-off value for the 5’ end of the transcripts was used in the analysis (how many base pairs, codons, constitutes the 5’end), is it the same length of 11 amino acids as Goodman et al. used?

13. Figure 1: The used color scheme is not color-blind friendly and indistinguishable in black & white. Since colors are referred to in the text, please use a colorblind friendly color scale (e.g. the viridis color scale for ggplot in R).

14. Please be consistent in referring to supplementary data (Supplementary Fig, S Fig, Fig S).

Reviewer #3: Lewin, Daniels and Hurst present an interesting concept of codon optimization for ribosomal initiation. Authors show that initiation optimal codons are different from translationally optimal codons with little or no overlap with each other. The findings are interesting and convey a clear message. However, the following concerns/questions should be answered prior to the publication.

(1) Translationally optimal codons are associated with the transcript with high translation efficiency. However, that is not the case with initiation optimal codons. Their presence perhaps slows down the initiation rate as they are mostly found in the genes with low protein abundance. Therefore, this one to one correspondence does not exist. Therefore, I am not sure if one should use the phrase initiation optimal codons.

(2) I believe that due to the unavailability of initiation rate, authors use protein abundance as a proxy of translation initiation rate. If the initiation rates are not available then I would recommend using mRNA copy number/protein copy number which is a better estimate of translation initiation rate.

(3) Authors find A/T ending codons promote high protein levels in E. coli. I think this result can be explained by low stability of mRNA structure near the start codon. Or there is something new here which I am missing.

(4) For the clarity of the readers, authors should discuss what they mean by the noise associated with the codon and how it is low at a lower initiation rate.

(5) Authors find that the transcripts that code for highly expressed proteins tend to have low initiation rate. Then, authors rationalize this observation by quoting previously published results that low initiation rate minimizes the noise. Another benefit of low initiation rate is that it minimizes the ribosome consumption by allowing a ribosome to produce proteins more efficiently (PMID: 37007710). Authors should discuss this additional benefit of low initiation rate in transcripts that code for highly expressed transcripts.

other comments:

(1) page 3, "a closely related approach employs codon ... employed overall". I think this is the codon harmonization approach which is employed to enhance the production of natively folded protein. This approach is used for proteins that tend to misfold. I don't think codon harmonization is used to enhance protein production.

(2) correlation in figure 2 is statistically significant. I would recommend them testing the correlation and p-value after removing one or two extreme points.

(3) Authors may consider shifting materials and methods before the results section.

(4) Introduction section is too long. It can be shortened.

**Have the authors made all data and (if applicable) computational code underlying the findings in their manuscript fully available?**

Reviewer #1: Yes

Reviewer #2: Yes

Reviewer #3: Yes

PLOS authors have the option to publish the peer review history of their article (what does this mean?). If published, this will include your full peer review and any attached files.

Reviewer #1: No

Reviewer #2: **Yes: **Charlotte C. Koster & John van der Oost

Reviewer #3: No
---

## [Decision Letter · Decision Letter 1]

6 Oct 2023

Dear Prof Hurst,

Thank you very much for submitting your manuscript "Genes for highly abundant proteins in Escherichia coli avoid 5’ codons that promote ribosomal initiation." for consideration at PLOS Computational Biology. As with all papers reviewed by the journal, your manuscript was reviewed by members of the editorial board and by several independent reviewers. The reviewers appreciated the attention to an important topic. Based on the reviews, we are likely to accept this manuscript for publication, providing that you modify the manuscript according to the review recommendations.

Reviewer 3 made a small suggestion, thus I'm sending you the mansucript under "Minor revision" to provide you the opportunity to make this change if you find it relevant, before final acceptance.

Sincerely,

Marc Robinson-Rechavi

Academic Editor

PLOS Computational Biology

Pedro Mendes

Section Editor

PLOS Computational Biology

Reviewer's Responses to Questions

**Comments to the Authors:**

Reviewer #1: My previous comments and questions have been adequately reponded and answered by the authors. I do not have further questions.

Reviewer #3: In the revised version, authors used the term high efficiency in the abstract. I think the authors meant the efficiency of ribosome usage. It should be clarified otherwise it gives the impression of translation efficiency which is different from the efficiency of ribosome utilization. All other questions are addressed by the authors.

**Have the authors made all data and (if applicable) computational code underlying the findings in their manuscript fully available?**

Reviewer #1: None

Reviewer #3: Yes

PLOS authors have the option to publish the peer review history of their article (what does this mean?). If published, this will include your full peer review and any attached files.

Reviewer #1: No

Reviewer #3: No

Figure Files:

Data Requirements:

Reproducibility:

References:

---

## [Editor Report · Decision Letter 2]

9 Oct 2023

Dear Prof Hurst,

We are pleased to inform you that your manuscript 'Genes for highly abundant proteins in Escherichia coli avoid 5’ codons that promote ribosomal initiation.' has been provisionally accepted for publication in PLOS Computational Biology.

Best regards,

Marc Robinson-Rechavi

Academic Editor

PLOS Computational Biology

Pedro Mendes

Section Editor

PLOS Computational Biology

---

## [Editor Report · Acceptance letter]

12 Oct 2023

PCOMPBIOL-D-23-01090R2 

Genes for highly abundant proteins in Escherichia coli avoid 5’ codons that promote ribosomal initiation.

Dear Dr Hurst,

I am pleased to inform you that your manuscript has been formally accepted for publication in PLOS Computational Biology. Your manuscript is now with our production department and you will be notified of the publication date in due course.

With kind regards,

Anita Estes
